# Fitness costs of female choosiness are low in a socially monogamous songbird

**Wolfgang Forstmeier**[1]*, **Daiping Wang**[1,2]*, **Katrin Martin**[1], **Bart Kempenaers**[1]

**1** Department of Behavioural Ecology and Evolutionary Genetics, Max Planck Institute for Ornithology, Seewiesen, Germany, **2** CAS Key Laboratory of Animal Ecology and Conservation Biology, Institute of Zoology, Chinese Academy of Sciences, Beijing, China

* forstmeier@orn.mpg.de (WF); wangdaiping@ioz.ac.cn (DW)

## Abstract

Female mate choice is thought to be responsible for the evolution of many extravagant male ornaments and displays, but the costs of being too selective may hinder the evolution of choosiness. Selection against choosiness may be particularly strong in socially monogamous mating systems, because females may end up without a partner and forego reproduction, especially when many females prefer the same few partners (frequency-dependent selection). Here, we quantify the fitness costs of having mating preferences that are difficult to satisfy, by manipulating the availability of preferred males. We capitalize on the recent discovery that female zebra finches (*Taeniopygia guttata*) prefer males of familiar song dialect. We measured female fitness in captive breeding colonies in which one-third of females were given ample opportunity to choose a mate of their preferred dialect (two-thirds of all males; "relaxed competition"), while two-thirds of the females had to compete over a limited pool of mates they preferred (one-third of all males; "high competition"). As expected, social pairings were strongly assortative with regard to song dialect. In the high-competition group, 26% of the females remained unpaired, yet they still obtained relatively high fitness by using brood parasitism as an alternative reproductive tactic. Another 31% of high-competition females paired disassortatively for song dialect. These females showed increased levels of extra-pair paternity, mostly with same-dialect males as sires, suggesting that preferences were not abolished after social pairing. However, females that paired disassortatively for song dialect did not have lower reproductive success. Overall, females in the high-competition group reached equal fitness to those that experienced relaxed competition. Our study suggests that alternative reproductive tactics such as egg dumping can help overcome the frequency-dependent costs of being selective in a monogamous mating system, thereby facilitating the evolution of female choosiness.

## Introduction

Whenever organisms face multiple options to choose from (e.g., choice of food, habitat, or mate), they have to weigh the potential benefits of being choosy against potential costs that arise from being too selective [1–8]. Over evolutionary timescales, the behavioral trait

**Data Availability Statement:** All underlying data can be found on the Open Science Framework under https://osf.io/6e8np.

**Funding:** This research was supported by the Max Planck Society (to BK). The funders had no role in

study design, data collection and analysis, decision to publish, or preparation of the manuscript.

**Competing interests:** The authors have declared that no competing interests exist.

"choosiness" may thus evolve to a fixed optimum level [9] or remain flexible depending on circumstances [10–14].

Female mate choice has been widely recognized as the driving force behind the evolution of many extravagant male ornaments and displays. Yet, whether such choosiness is expected to evolve should depend critically on how costly it is to be choosy [15–17]. The costs of choosiness are hence central to sexual selection theory, but they have rarely been measured empirically (see below). The costs of being selective about a mate as opposed to mating with the first potential mate that is encountered will greatly depend on the species' mating system.

Some of the most spectacular examples of sexually selected display traits have been observed in lek mating systems with strong reproductive skew, i.e., systems in which most or even all females in a given area can mate with the same male (e.g., black grouse, *Lyrurus tetrix* [18] and capuchinbird, *Perissocephalus tricolor* [19]). In general, females can mate with the same male if they do not seek a partner who provides nonshareable direct benefits (e.g., parental care), but only mate to obtain sperm (i.e., genetic benefits), provided sperm depletion is not an issue. Intense selection through female choice for the most attractive males should, however, erode genetic variation, which will then reduce the genetic benefits that females can obtain from being choosy. The apparently remaining female choosiness in face of diminishing benefits is widely known as the "paradox of the lek," which has been addressed in numerous theoretical and empirical studies [20]. The empirical work has concentrated on quantifying (a) the costs to females of being choosy in terms of time and energy spent or in terms of predation risk [21–24]; and (b) the magnitude of genetic benefits from mating with the preferred male [25]. When the costs and benefits are measured on a relevant and comparable scale, i.e., in terms of fitness consequences for the female, they appear to be so small that they can hardly be quantified with sufficient precision to provide an empirical answer to the lek paradox [26,27].

Monogamous mating systems should provide a more tractable opportunity to study the evolution of choosiness empirically, because both costs and benefits of choosiness should be much larger than in lek mating systems. In socially monogamous systems, males typically provide substantial direct benefits in the form of parental care. If the quality or quantity of parental care varies among males, females may obtain large fitness gains from selecting the best partner available [9,28–30]. Hence, females may have more to gain from being choosy (compared to those in a lekking system), provided that they can reliably identify males that provide larger benefits (e.g., a "good parent" [16,28]). However, a female that is too selective might not find any partner that satisfies her choice criteria ("wallflower effect" [31]), especially because the best partners will rapidly disappear from the available mating pool, and thereby risk having to raise offspring without male help. Thus, strong female competition over the best mates may lead to selection against being too choosy [15] and hence favor strategies such as accepting the first mate encountered if its quality lies within the top 80% of the males (i.e., only discriminating against the bottom 20%). Yet, such theoretical predictions about optimal female choosiness should critically depend on behavioral tactics that females can adopt when their preferences cannot be satisfied and on the fitness consequences of these tactics (Fig 1). This choice of tactics can be studied empirically, but we are not aware of any systematic work on this topic despite its central importance for sexual selection theory.

When many females compete for a limited number of preferred partners, they pay a cost of engaging in competition (time and energy spent in competition, risk of injury), compared to females that are not constrained by their preferences, either because their preferred partners are overabundant or because they are not choosy (Fig 1: "cost of competition"). The cost of competition can be equal for both winners and losers of the competition (as in Fig 1), but females might also vary in their abilities to avoid this cost (e.g., by "prudent mate choice" [12]). Females that are unsuccessful at securing a preferred partner can either settle for a partner

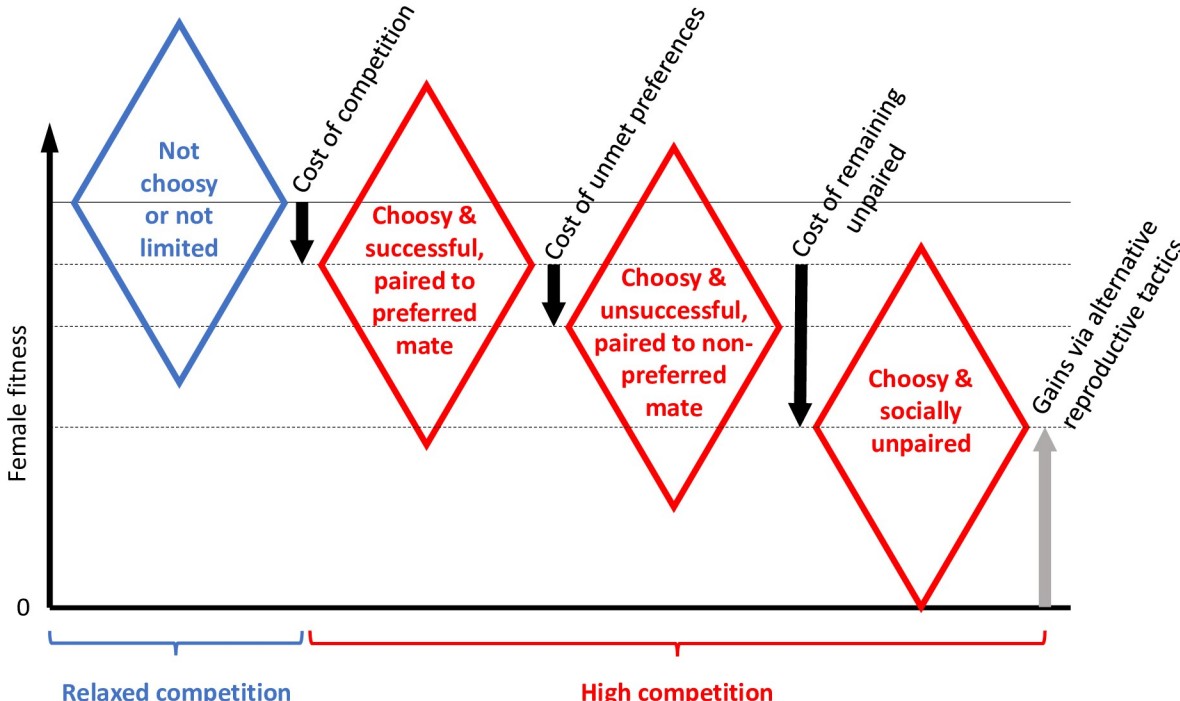

**Fig 1. Schematic representation of the expected fitness costs of choosiness.** Costs of choosiness for females that are limited by the availability of preferred mates (red, high competition) compared to females that are not limited by their choosiness or by the availability of preferred mates (blue, relaxed competition). For simplicity, we assume that preferred and nonpreferred mates do not differentially affect female fitness. Diamonds illustrate variation in individual fitness around the mean fitness of a group of females (center of diamonds, horizontal lines). Black arrows represent various aspects of costs of choosiness under competition for mates. The gray arrow indicates fitness gains via alternative reproductive tactics for females that remain socially unpaired, including reproduction as single female or via parasitic egg dumping. The cost of unmet preferences may, for example, result from reduced willingness to copulate leading to infertility, aggression, and reduced male brood care. Note that in empirical studies, the apparent cost of unmet preferences and the cost of remaining unpaired might be confounded by effects of intrinsic quality differences between the 3 groups of females shown in red. Also note that all choosy females (red) pay a cost of competition, which might also vary between groups, for example, if some females avoid direct competition.

they do not prefer or remain socially unpaired. Females that settle for a partner they do not prefer may have to pay 2 types of costs. First, their partner may be of low quality and hence may provide less benefits than the average partner. Note that we have omitted this quantity from Fig 1, because it is equivalent to how much there is to be gained from being choosy (benefits of choice). Here, we focus only on the costs of choosiness in the absence of variation in benefits (i.e., all males are equally good parents, as is the case in the empirical study reported here). Second, even when all males are of equal quality, females paired to a partner they did not prefer may still suffer a "psychological" cost that arises from the preferences not being met (Fig 1: "cost of unmet preferences"; see [30]). For example, females may be more reluctant to copulate with their partner, resulting in infertility, or they may prefer to copulate with males outside the pair bond, which may lead to aggression [32] and reduced parental care by the social partner [33]. Hence, even if females would be able to satisfy their preferences via extrapair copulations, they may still pay a cost when extra-pair mating has negative effects on cooperation with the social partner. One way to avoid such costs might be to behave similarly toward a preferred and a nonpreferred partner once paired. Finally, in cases where females remain socially unpaired, they will also pay a cost (Fig 1: "cost of remaining unpaired"), the magnitude of which will depend on how successfully females can achieve fitness through

alternative reproductive tactics, including reproduction as a single mother [34] or via brood parasitism ("egg dumping" [35–37]).

Mate choice in socially monogamous mating systems (and in other competitive systems [38]) has the intriguing property that selection on mating preferences works in a negative frequency-dependent manner [17,39–41]. Generally, negative frequency-dependent selection means that the fitness of a variant decreases as it becomes more frequent. In the context of mate choice, expressing preferences may bear little costs when the availability of preferred mates is unlimited relative to demand. However, in a socially monogamous system, preferred mates may be a highly contested resource, particularly when most competitors prefer the same kind of mates. Hence, the fitness consequences of an individual's preferences depend on what other individuals in the population prefer, and the larger the number of competitors with the same preferences, the stronger the competition for the same few mates. For example, if two-thirds of all females would only accept a partner that ranks in the top third of all males (e.g., with regard to ornament size), then at least half of those females will remain unpaired, thereby lowering the mean fitness of all females that carry such preference alleles. As a consequence, such preferences can be strongly selected against, particularly when a male ornament is a poor indicator of benefits to the female [29,41,42]. Selection against such preferences will be strongest when the preferences are shared by most females, and negative frequency dependence should ultimately result either in the loss of preference or in diversification of preferences, leading to relatively little consensus among females about which male is the most attractive [40,41,43,44].

To understand the evolution of optimal levels of choosiness in monogamous mating systems, it is essential to quantify empirically the fitness costs of having preferences that are difficult to satisfy. Although several studies have manipulated the costs and/or benefits of choosiness and have subsequently observed female choice behavior or mating patterns [45–51], no study to date has quantified the costs in terms of female fitness. Only measurements of female fitness allow us to judge the strength of selection on mate preferences. Fitness should thereby be measured in a natural (or at least naturalistic) setup that allows the expression of all existing forms of behavioral plasticity that may have evolved to reduce the costs of having preferences that are difficult to satisfy.

One practical obstacle is that the costs of choosiness can only be measured if one finds a sufficiently strong preference that will be reliably expressed by the choosing sex. In zebra finches, a socially monogamous bird that forms lifelong pair bonds, females reliably prefer (unfamiliar) males that have learned their song in the same population in which females grew up, over males with song from a different population [52]. Working with 4 independent captive populations (2 domesticated and 2 recently wild derived), we used cross-fostering of eggs between populations to produce 2 different cultural lineages (A and B) within each population that differ only in their song dialects. The lineages were bred in isolation for one additional generation, to obtain birds from the same genetic population that differ only in the song that the foster grandparents once transmitted to the parents of the current generation. When bringing together equal numbers of unfamiliar males and females of the 2 song dialects A and B, on average, 73% of pairs formed assortatively by dialect (random expectation: 50% [52]). We made use of this moderately strong assortative mating preference to design an experimental study with preregistered methods of data collection and analysis plan (https://osf.io/8md3h), ensuring maximal objectivity in the quantification of fitness costs of choosiness.

We set up a total of 10 experimental aviaries (2 or 3 per genetic population). In each aviary, we placed 12 males and 12 females from lineages A and B in a 2:1 or 1:2 ratio (e.g., 4 females of lineage A and 8 females of B, facing 8 males of A and 4 males of B). In this way, we created groups of females that have either plenty of preferred males to choose from ("relaxed

competition") or that have to compete for a limited pool of preferred mates ("high competition"). The latter group can thus accept a nonpreferred mate (i.e., mate disassortatively) or forego forming a pair to reproduce (Fig 1). This design mimics the above-described example of a two-thirds majority preferring a male from the top third, while the other group of females are nearly unconstrained by their preference, and it mirrors the principle of negative frequency dependence of preferences in a monogamous system. The treatment thus alters the cost of preferring the same lineage, while the benefits of having that preference should equal 0 for both treatment groups (as we assumed in Fig 1, which otherwise can be adapted to accommodate variation in benefits).

Note that our experiment did not manipulate female choosiness. Rather, we examined the consequences of naturally occurring levels of choosiness. All females are assumed to have equally strong dialect preferences, and we measured the costs of having such preferences under 2 conditions of availability of preferred males (nonlimiting versus. limiting). These 2 conditions reflect the principles of frequency-dependent selection, namely that the costs of preferences should be large when many females compete for the same few mates that they prefer and small or absent when preferred males are abundant. We quantified the full fitness consequences of the treatment, including all consequences of behavioral plasticity in both sexes (e.g., egg dumping, extra-pair mating, and the effects of the response by the social partner). The experimental design allowed us to rule out fitness variation due to variation in the benefits of choice arising from differences in male quality, because the males of preferred and nonpreferred song dialect are of equal quality. Hence, we can measure the sum of all costs arising from the limited availability of preferred mates while keeping all benefits of choice constant.

We allowed all birds to reproduce freely for a fixed period (70 days for egg laying plus 50 days for chick rearing) and quantified the fitness costs of choosiness, closely adhering to the preregistered plan (https://osf.io/8md3h). Prior to data collection, we had hypothesized that (1) females from the high-competition treatment will achieve lower relative fitness (measured as the number of independent offspring; primary outcome) compared to the females from the relaxed-competition treatment. Further, we hypothesized that these females (2) will lay fewer eggs; and (3) will start egg laying later (secondary outcome measures). We further present the results of an unplanned, exploratory data analysis to elucidate mechanisms by which females coped with the experimental challenge (see Fig 1).

## Results

### A. Preregistered analyses: Costs of choosiness

The 120 experimental females produced a total of 556 offspring that reached independence (mean offspring per female ± SD = 4.6 ± 3.0, range 0 to 13). As expected, relative fitness of females decreased with their inbreeding coefficient (mean F ± SD = 0.051 ± 0.050, range: 0 to 0.28; $p$ = 0.006, S1 Table, Model 1a). However, in contrast to our a priori prediction, the 80 females in the high-competition treatment achieved a nonsignificantly higher (rather than lower) relative fitness (1.022 ± 0.069) compared to the 40 females in the relaxed-competition treatment (0.955 ± 0.097; $p$ = 0.57; Fig 2, Table 1, Model 1a, S1 Table). This result did not change after additionally controlling for additive genetic and early environmental effects on fitness (S1 Table, Model 1b). Moreover, and also in contrast to our predictions, females from the high-competition treatment did not lay fewer eggs (9.2 ± 0.4) than females from the relaxed-competition group (8.6 ± 0.6; $p$ = 0.37; Table 1, Model 2, S2 Table), and they did not start egg laying later (back-transformed means, high competition: 7.8 days after the start of the experiment, interquartile range of raw data: 5 to 10.5 days; relaxed competition: 8.2 days, interquartile range: 5 to 10.5 days; $p$ = 0.63; Table 1, Model 3, S3 Table).

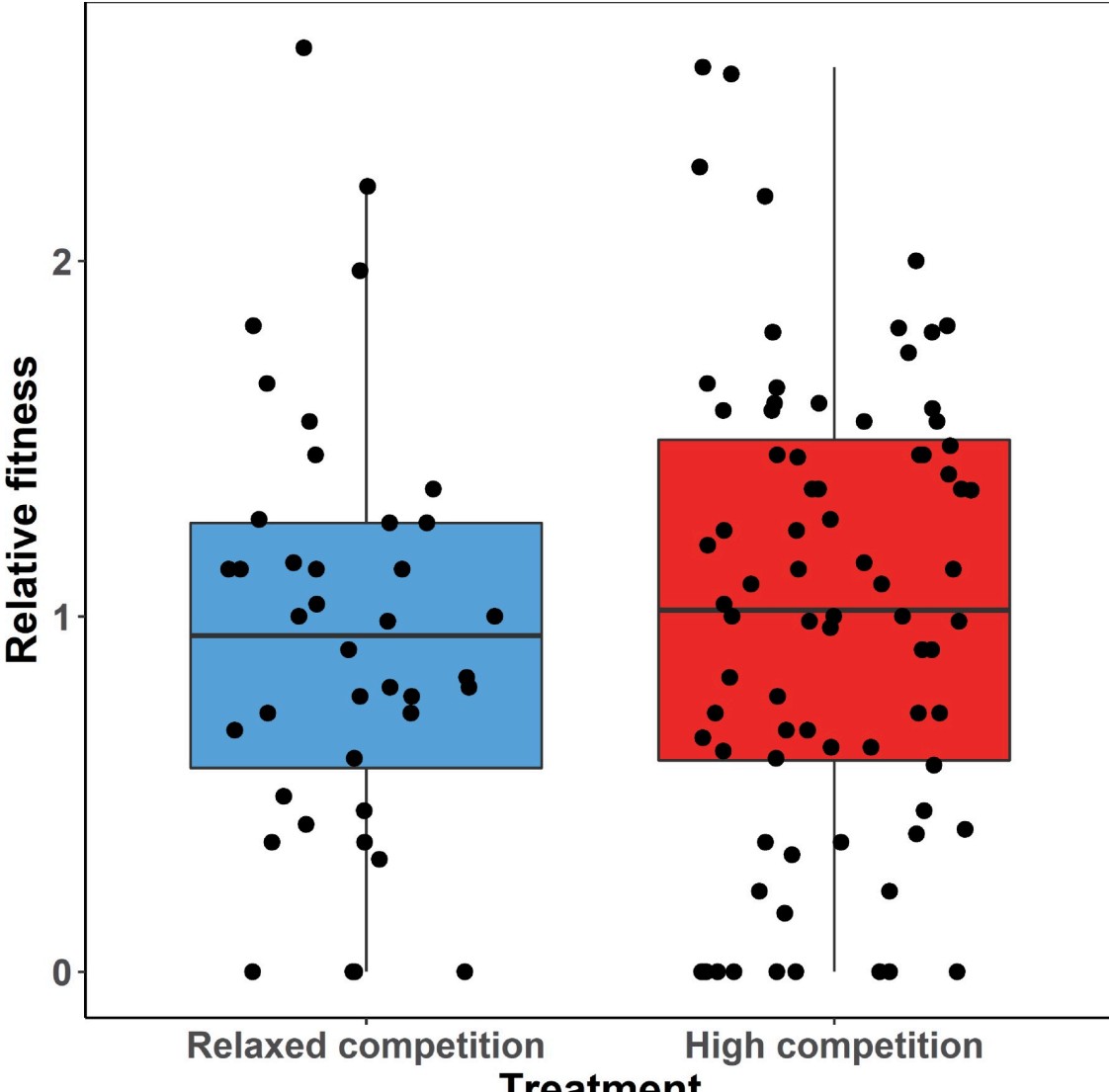

**Fig 2. Observed relative fitness of females from the 2 treatment groups.** Relative fitness is measured as the number of independent offspring produced by each female, scaled to a mean of one within each of 10 experimental aviaries. Shown are fitness values of the 40 females from the relaxed-competition group (4 females with 8 males of their preferred natal song dialect per aviary) and of the 80 females from the high-competition group (8 females with 4 males of their preferred natal song dialect per aviary). Dots represent individual females and are jittered horizontally to increase visibility. The box plot indicates group medians (0.95 and 1.02) and interquartile ranges (25th and 75th percentiles). Whiskers show the data range except for "outliers" (defined as laying beyond 1.5 times the interquartile range above or below the 25th and 75th percentiles). The data underlying this figure can be found in https://osf.io/6e8np/.

## B. Post hoc data exploration: Female coping tactics

The lack of significant treatment effects could be due either to a failed treatment (e.g., because birds did not prefer their natal song dialect) or to female behavior that avoids costs of choosiness. Hence, we first examined the efficiency of the treatment, i.e., the degree of assortative mating by song dialect. Second, we investigated the mechanisms by which females reproduced, i.e., we compared success and timing of social pairing, alternative reproductive tactics, and rearing success between the 2 treatment groups.

**Table 1. Comparisons between females of the "high-competition" (*n* = 80) and "relaxed-competition" (*n* = 40) treatment.**

| Model | Test type | Dependent variable | High competition | Relaxed competition | *p* (treatment) | Trend in expected direction | Covariates | Random effects |
|-------|-----------|--------------------|------------------|---------------------|-----------------|-----------------------------|------------|----------------|
| 1a | Planned | Relative fitness (scaled to unity) | 1.023 | 0.953 | 0.57 | No | F | - |
| 1b | Planned | Relative fitness (scaled to unity) | 1.023 | 0.953 | 0.32 | No | F, peer size, mother fitness | - |
| 2 | Planned | N genetic eggs laid | 9.21 | 8.58 | 0.37 | No | F | Exp AV, natal AV |
| 3 | Planned | Latency to first genetic egg (days) | 7.78 | 8.25 | 0.63 | No | F | Exp AV, natal AV |
| 4 | Exploration | Proportion females socially unpaired | 26% | 10% | 0.064 | Yes | F | Exp AV, natal AV |
| 5 | Exploration | N social bonds per female | 0.863 | 0.925 | 0.42 | Yes | F | Exp AV, natal AV |
| 6 | Exploration | N assortative social bonds | 0.450 | 0.900 | 0.000002 | Yes | F | Exp AV, natal AV |
| 7 | Exploration | N disassortative social bonds | 0.413 | 0.025 | 0.0014 | Yes | F | Exp AV, natal AV |
| 8 | Exploration | Latency to first social bond with eggs (days) | 13.48 | 7.93 | 0.008 | Yes | F | Exp AV, natal AV |
| 9 | Exploration | N clutches attended as a single mother | 0.163 | 0.125 | 0.57 | Yes | F | Exp AV, natal AV |
| 10 | Exploration | N eggs actively taken care off | 6.64 | 7.40 | 0.21 | Yes | F | Exp AV, natal AV |
| 11 | Exploration | N eggs dumped to other females (strict) | 1.63 | 0.80 | 0.038 | Yes | F | Exp AV, natal AV |
| 12 | Exploration | N eggs dumped anywhere (wide) | 2.58 | 1.18 | 0.009 | Yes | F | Exp AV, natal AV |
| 13 | Exploration | Proportion eggs dumped (strict) | 17% | 9% | 0.029 | Yes | F | Exp AV, natal AV, FID |
| 14 | Exploration | Proportion eggs dumped (wide) | 29% | 12% | 0.002 | Yes | F | Exp AV, natal AV, FID |
| 15 | Exploration | Proportion fertile eggs leading to offspring | 50.1% | 50.2% | 0.88 | Yes | F | Exp AV, natal AV, FID |

Overview of planned tests (Models 1 to 3, as outlined in the preregistration document before data collection; https://osf.io/8md3h) and post hoc tests that were conducted after knowing the results of the planned tests (data exploration, Models 4 to 15). All conducted tests are reported in their initial form (no selective reporting, no post hoc modification). Indicated are average values for the 2 treatment groups for each dependent variable. Proportions of eggs refer to means of individual mean proportions. For latencies, back-transformed values after averaging $\log_{10}$-transformed values are shown. *p*-Values refer to group differences based on glms or glmms. Covariates are the female's inbreeding coefficient (F), the size of the peer group in the female's natal AV (peer size), and the fitness of the female's mother. Random effects are the exp AV (10 levels), the female's natal AV (16 levels), and—in binomial models of counts with overdispersion—FID (120 levels) (see S1–S15 Tables for details). Note that the high significance of the treatment effect in Models 6 and 7 is partly caused by the experimental design.

exp AV, experimental aviary; FID, female identity; natal AV, natal aviary.

The degree of assortative mating, i.e., the proportion of assortative pairs, can range from 0 to 1 (see Fig 3). In our experimental setup, a value of 0 can theoretically be reached if all pairs mated disassortatively (0 assortative and up to 12 disassortative pairs in each aviary). A value of 1, corresponding to perfect assortative pairing, can only be reached if 4 females per aviary remained unpaired (8 assortative and 0 disassortative pairs per aviary). Under random pairing, 44.4% of pairings should be assortative (1/3 of females has a 2/3 chance of pairing assortatively, plus 2/3 of females has a 1/3 probability; 1/3 × 2/3 + 2/3 × 1/3 = 4/9). If all females would attempt to pair assortatively, but no female would forego pairing, 66.7% of pairings should be assortative (8 assortative and 4 disassortative pairs per aviary).

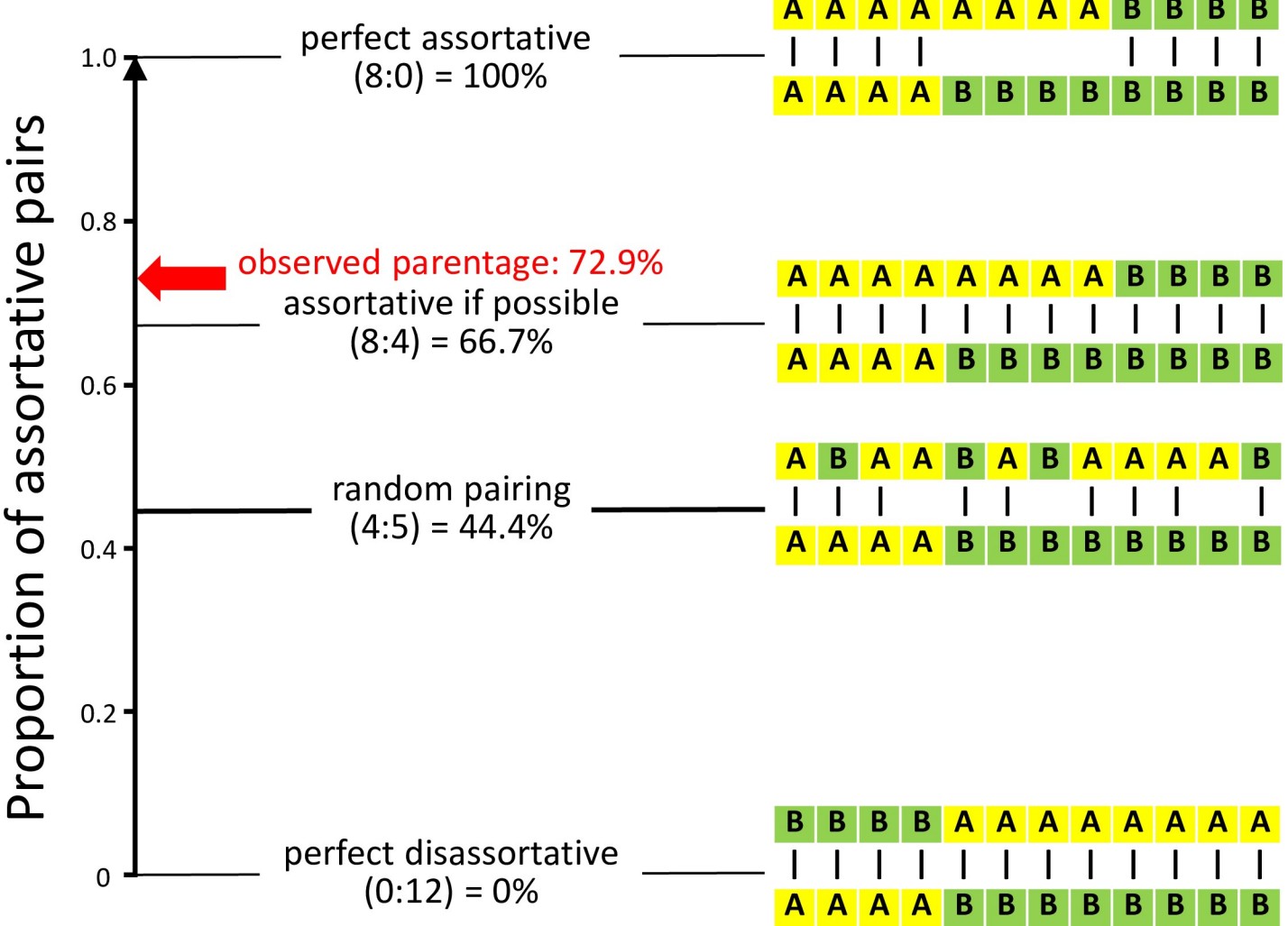

**Fig 3. Expected and observed levels of assortative mating under the given experimental design.** Letters A and B stand for individuals of different song dialects in an aviary (each row represents one sex) and dashes connecting letters represent pair bonds resulting in different levels of assortative mating with regard to song dialect. Random pairing on average produces 44.4% assortative pairs (pairs matched for their song dialect). "Observed parentage" refers to the proportion of fertilized eggs ($N = 1,074$) of which the genetic parents were mated assortatively. For comparison, 4 idealized scenarios of pairing are indicated together with the numbers of assortative versus disassortative pairs (in parentheses). The data underlying this figure can be found in https://osf.io/6e8np/.

Of the 106 social pairs that were observed (involving 95 different males and 95 different females), 72 (67.9%) were assortative. This significantly deviates from the random expectation of 44.4% (exact goodness of fit test $p < 0.0001$). Considering the number of eggs in the nests of those pairs ($N = 1,022$ in total), 730 eggs (71.4%) were cared for by assortative social pairs. At the level of fertilization, out of the 1,074 eggs fertilized and genotyped, 783 (72.9%) had parents of the same song dialect (females of relaxed-competition treatment: 325 out of 342 eggs, 95.0%; high competition: 458 out of 732 eggs, 62.6%). Hence, both at the social and genetic level, we found strong assortative mating, slightly exceeding the 66.7% "assortative if possible" threshold (Fig 3).

Females from the 2 treatment groups differed in their pairing success, with only 4 females (10%) from the relaxed-competition treatment remaining unpaired, but 21 females (26%) from the high-competition treatment not observed in a social pair bond ($p = 0.064$, Fig 4A and 4B, Table 1, Model 4, S4 Table). In the relaxed-competition group, 87.5% of females ($N = 35$)

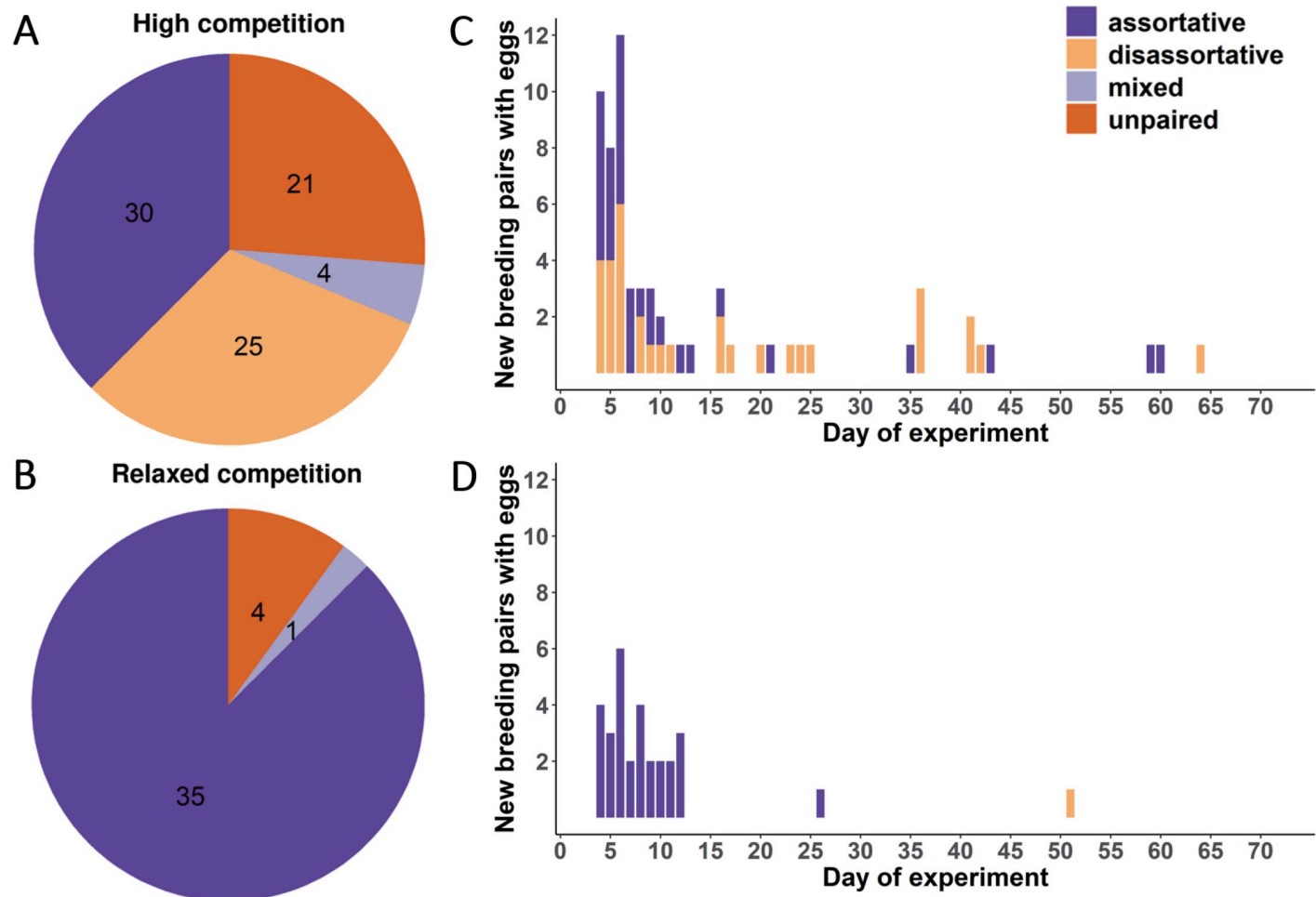

**Fig 4. Observed pair bonds for females from the relaxed and high-competition groups. (A and B)** Pie charts showing the proportion of females in each of the 2 treatment groups that were either not observed as a pair (unpaired) or were seen in assortative, disassortative, or both type of pair bonds (mixed). Numbers indicate the count of females in each group. **(C and D)** Histograms illustrating the temporal patterns of emergence of social bonds (either assortative or disassortative). Shown is the day after the start of the experiment (potentially ranging from 1 to 70) on which the first egg was recorded in a nest taken care of by one of the 106 breeding pairs (note that this may include parasitic eggs not laid by the focal female). Note that assortative bonds ($N = 72$) formed significantly earlier than disassortative bonds ($N = 34$; back-transformed estimates 9.3 versus 17.8 days, $t$ test on log-transformed latency: $t_{104} = 3.67$, $p = 0.0004$). The data underlying this figure can be found in https://osf.io/6e8np/.

mated assortatively with a male from their natal dialect, and 1 female (2.5%) was observed in 2 pair bonds (1 assortative, 1 disassortative; "mixed" in Fig 4B). By contrast, in the high-competition group, only 37.5% of females ($N = 30$) mated exclusively assortatively, 5% ($N = 4$ females) participated in both types of pairing, and 31% ($N = 25$ females) mated exclusively disassortatively (Fig 4A, Table 1, Models 5–7, S5–S7 Tables).

Females from the high-competition group took longer to start a social bond compared to females from the relaxed-competition group ($p = 0.008$, Fig 4C and 4D, Table 1, Model 8, S8 Table). For this test, we assigned a maximum latency of 75 days to unpaired females (as in the preregistered Model 3), because we cannot exclude that such females would have paired after a longer period. If unpaired females ($n = 25$) are excluded from the analysis, the difference between treatment groups in latency to pair is no longer significant ($t_{93} = 1.24$, $p = 0.22$). However, a Cox proportional hazard model that includes all females (Model 8a, S17 Table, S1 Fig) shows that the treatment significantly delayed social pairing in the high-competition group

relative to the relaxed-competition group (hazard ratio = 0.618, $p$ = 0.025). Hence, the treatment prevented or delayed social pairing (Table 1, Models 4 and 8a, S4 and S17 Tables), but it did not prevent or delay egg laying (S2 and S3 Tables).

In 18 cases, females attempted to rear offspring as single mothers (14 females attended 1 clutch and 2 females each attended 2 consecutive clutches). Of those, 11 clutches (61%) were reared by females that remained unpaired until the end of the experiment (overall, 25 out of 120 females remained unpaired until the end, 21%). However, the average number of clutches attended to as unpaired female did not differ significantly between the treatment groups ($p$ = 0.57, Table 1, Model 9, S9 Table). Females from the high-competition group on average laid fewer eggs that they actively took care of, although this was not significant ($p$ = 0.21, Table 1, Model 10, S10 Table). However, females from the high-competition group laid significantly more eggs into clutches that were cared for by other females (egg dumping in the strict sense; $p$ = 0.038, Table 1, Model 11, S11 Table) and into nests of other females, nests attended by single males, or into unattended nest boxes (egg dumping in the wide sense; $p$ = 0.009, Table 1, Model 12, S12 Table). Hence, the proportion of parasitic eggs among the total number of eggs laid was markedly higher in the high-competition than in the relaxed-competition group (Table 1, Models 13 and 14, S13 and S14 Tables).

Splitting the females of each treatment group into subsets according to their social pairing status (Fig 4) shows that the parasitic egg dumping tactic was used more often by the unpaired females of the high-competition group (compared to all females in the relaxed-competition group; $t$ test with unequal variances, $t_{26.4}$ = 3.37, $p$ = 0.002), followed by the disassortatively mated females of the high-competition group (again compared to all females in the relaxed-competition group $t_{33.4}$ = 2.09, $p$ = 0.044; Fig 5B). Overall, females from the high-competition group achieved similar fitness to the females from the relaxed-competition group (Fig 5A), because rearing success (the proportion of fertile eggs that became independent offspring) did not differ between the treatment groups ($p$ = 0.87, Table 1, Model 15, S15 Table). Note that embryo and nestling mortality affected about 50% of fertilized eggs (Table 1), which is typical for these captive populations [53].

Finally, we examined levels of extra-pair paternity in the different treatment groups, focusing on the 84 females that were socially paired to only one partner. As expected, extra-pair paternity was more frequent in the disassortatively paired females from the high-competition group (44%, 81 out of 183 eggs from 21 females) than in the assortatively paired females from the same treatment group (18%, 42 of 252 eggs from 27 females; $t$ test based on proportions for each female: $t_{46}$ = 3.2, $p$ = 0.002; Fig 6). Assortatively paired females from the relaxed-competition group showed intermediate levels of extra-pair paternity (36%, 112 of 312 eggs from 35 females; for additional details, see S16 Table). In each of the 3 groups, the majority of extra-pair eggs were sired assortatively (70%, 65%, and 89%, respectively) and all 3 numbers clearly exceed the corresponding random expectations (36%, 27%, and 64%, respectively) calculated from the number of potential extra-pair males in the aviary (4, 3, and 7 out of 11, respectively; see also S16 Table).

## Discussion

Our study illustrates the importance of empirically quantifying the costs and benefits of choosiness to predict selection on the level of choosiness, which can then inform discussions about the expected intensity of sexual selection through female choice. A recent theoretical study highlighted that choosiness in monogamous systems may have high costs and hence will be selected against [15]. Based on this study, we hypothesized that females in the high-competition group would suffer substantial fitness costs compared to those in the relaxed-competition group (https://osf.io/8md3h). However, our empirical findings strongly suggest that female

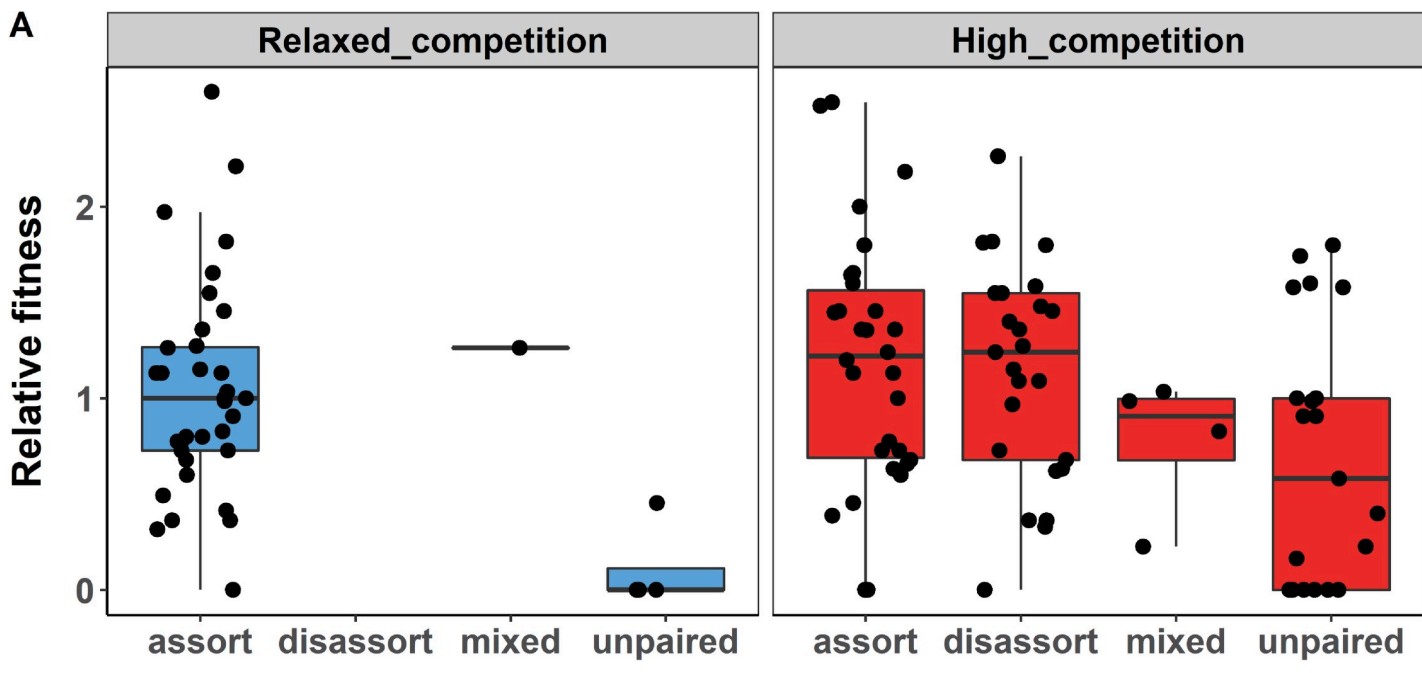

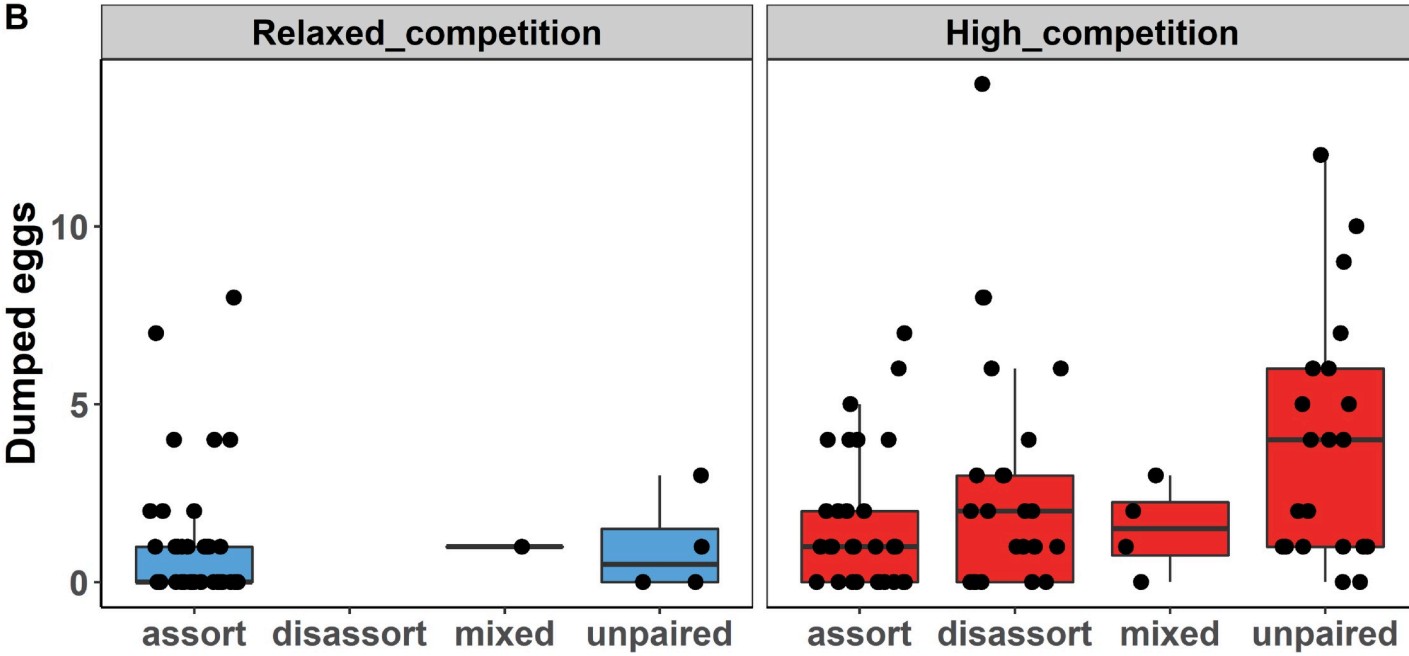

**Fig 5. Fitness and brood parasitism as a function of pairing status.** (A) Relative fitness, as described in Fig 2 and (B) number of genetically verified "dumped eggs" (broad definition of parasitic eggs) for females of different pairing status (unpaired, or mated assortatively, disassortatively or both, as in Fig 4A and 4B) in each of the 2 treatment groups (relaxed competition versus high competition). Horizontal lines indicate group medians (for other details, see legend of Fig 2). The data underlying this figure can be found in https://osf.io/6e8np/.

zebra finches have evolved sufficient behavioral flexibility to cope with the challenge of having preferences that are difficult to satisfy, such that they did not suffer lower fitness. This flexibility is not trivial, because zebra finches that were force paired suffered significant fitness costs compared to birds that were allowed to choose their mate [30].

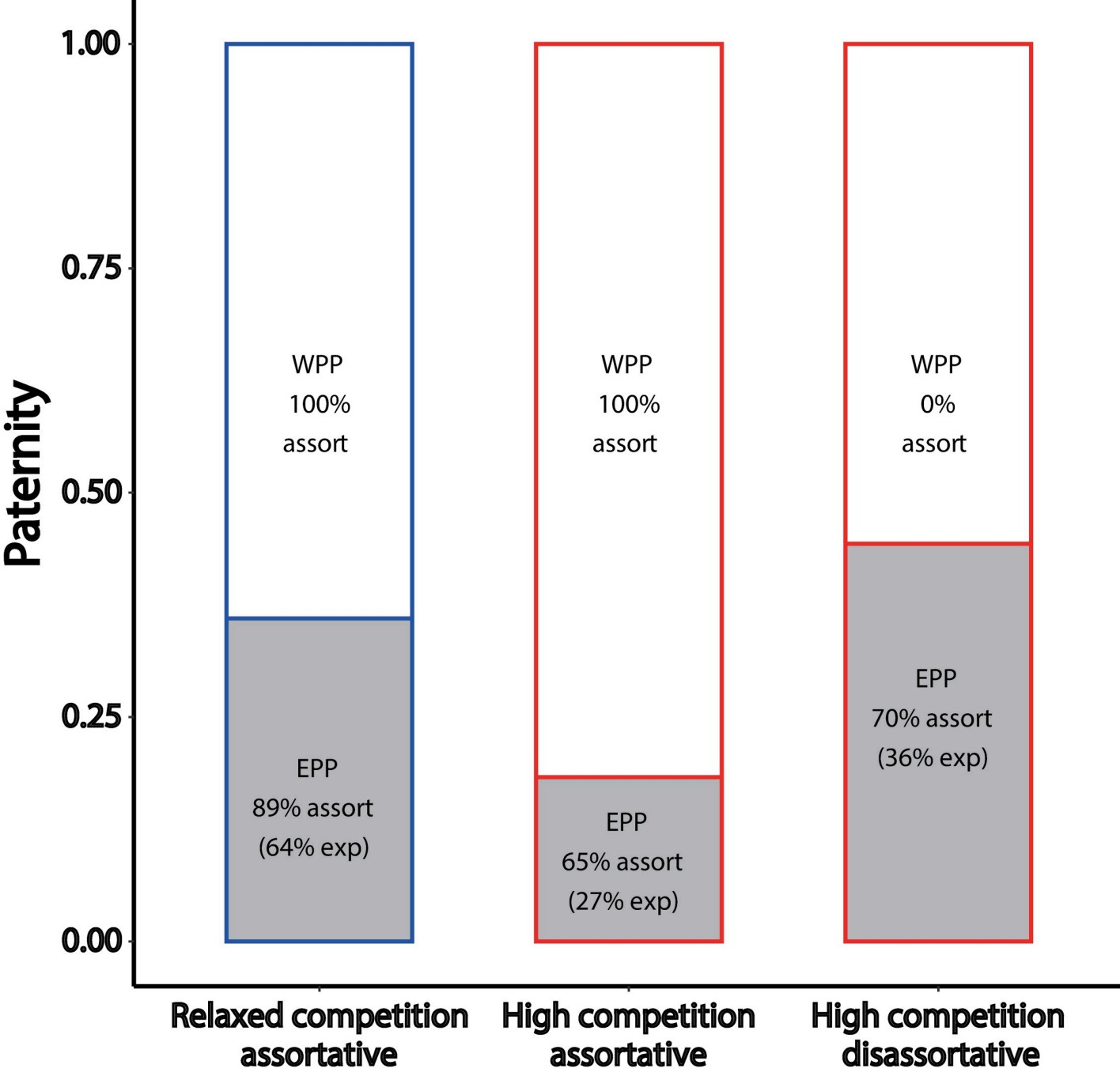

**Fig 6. Extra-pair mating as a function of pairing status.** Proportion of eggs sired outside the monogamous pair bond EPP (gray bars) versus WPP (white bars) for 3 groups of females with a single social pair bond. These are (1) assortatively paired females (*n* = 35) from the relaxed-competition treatment (blue); (2) assortatively paired females (*n* = 28, one of which did not lay any eggs) from the high-competition treatment (red); and (3) disassortatively paired females (*n* = 21) from the high-competition treatment. For each category of eggs, we indicate the proportion that is sired assortatively ("assort") for song dialect and in parentheses the random expectations ("exp") for this proportion of assortative mating based on the number of available extra-pair males of each song dialect. For more details, see also S16 Table. The data underlying this figure can be found in https://osf.io/6e8np/. EPP, extra-pair paternity; WPP, within-pair paternity.

Females in the high-competition treatment on average achieved slightly higher relative fitness compared to those in the relaxed-competition group. This difference was even more pronounced (yet still nonsignificant, $p = 0.32$) after accounting for possible confounding factors,

such as heritable variation in female fitness and variation in rearing conditions (compare Model 1b to Model 1a in S1 Table). These results are incompatible with our starting hypothesis of a substantial cost of being choosy when the availability of preferred mates is limited. Thus, the best estimate for the fitness cost of choosiness in our study equals 0. However, when considering reduced pairing success, delayed pairing and reliance on conditional parasitism, one could argue that the biologically most likely fitness cost is small, but positive. Females that relied on the parasitic tactic of egg dumping [35–37] were surprisingly successful in terms of fitness (Fig 5). However, our models on the use of this tactic also suggest that this may be a form of "making the best of a bad job," because the proportion of a female's eggs that was dumped (in the wide sense) rather than actively cared for was higher in the high-competition treatment group and also increased with the inbreeding coefficient of the female ($p = 0.002$, S14 Table). These results suggest that the parasitic tactic is associated with poor pairing success and with poor female condition. Hence, overall, there likely is a small net cost of having preferences that are hard to satisfy, but quantifying such a small cost is difficult because of sampling noise.

Our study suggests that an alternative reproductive tactic, namely egg dumping, may be important to consider as a mechanism that effectively reduces the costs of choosiness and thereby favors the evolution of choosiness even in monogamous mating systems. Alternative reproductive tactics can thereby increase the intensity of sexual selection through female choice. Rates of egg dumping reported for zebra finches breeding in the wild range from 5% to 11% of the eggs [54, 55], and a similar rate (6%) has been found in one of our captive populations [36]. Females of the relaxed-competition group showed a comparable rate of egg dumping (9% of eggs, using the strict definition comparable to the definitions used in those previous studies), while a considerably higher rate (17%) was observed in the females of the high-competition treatment. Note that our analysis of egg dumping is part of the post hoc data exploration rather than preregistered hypothesis testing, which implies that the probability that this result is a chance finding is higher (Fig 5). Nevertheless, we avoided extensive exploratory testing combined with selective reporting and post hoc modification of analysis strategy to minimize the risks of false-positive findings [56]. Accordingly, Table 1 presents all the exploratory tests that compare the 2 treatment groups in their original version. The considered hypotheses all directly follow from the observation of equal fitness in both treatment groups and address the question how females in the high-competition group responded to the given mating opportunities.

This study also contributes to our understanding of zebra finch mating preferences with regard to song dialects. Firstly, we confirmed that such preferences exist and that they are sufficiently strong to result in a high degree of assortment even when bringing together unequal numbers of males and females of each dialect. Secondly, assortative mating was present both at the level of realized fertilization (72.9% of fertilized eggs had assortative genetic parents) and at the level of social relationships (71.4% of eggs were cared for by assortative pairs). As 78.2% of all eggs sired by extra-pair males were assortative, we infer that song dialect preferences affect both social pairing and extra-pair mate choice.

The possible adaptive function of these preferences in the wild is not known. They could function to enhance local mating to obtain locally adapted genes, which would be adaptive in both the social and extra-pair context. However, this possibility seems unlikely in light of the lack of genetic differentiation even over a large geographic distance [57]. More widespread sampling of genotypes throughout Australia would be required to rule out this possibility. Alternatively, song preferences could function to find a mate that hatched locally [58,59] and hence may have gathered local information on ecologically relevant factors such as resources and predation risk. In that case, the song preferences during extra-pair mating might represent

a (nonadaptive) spillover of preferences that are functional in social pairing, or extra-pair mating could function to maintain additional social bonds with similar direct benefits [60].

Our study was designed to estimate the costs of female choosiness. We predicted that this cost would be high in a monogamous mating system with biparental care [15]. However, this is not what we found. Although our experimental treatment was effective in eliciting strong assortative mating preferences (Figs 2 and 3), females avoided substantial fitness costs under high competition for preferred males, at least in our aviary setting. Thus, our study does not support the hypothesis that female choosiness is costly in a socially monogamous system. Females from the high-competition treatment were affected in terms of delayed pairing and reduced pairing success, but they made up for this primarily by using the alternative reproductive tactic of egg dumping and only rarely by caring for clutches as a single mother. Females who ended up paired with a nonpreferred partner were more likely to engage in extra-pair copulations, but this did not affect their fitness (see Fig 5). This stands in contrast to an earlier study that showed that force-paired females responded more negatively to courtships by their social partner, had reduced fertility, and received less paternal care, resulting in a significant reduction in fitness [30]. The difference in treatment effects may be explained by the fact that, in the present study, all females could still choose their partners, while in the previous study, females were force paired, which may have resulted in behavioral incompatibility of partners. Overall, our results emphasize that models of the costs of choosiness need to be informed by empirical research.

## Methods

All methods closely adhere to the preregistration document (https://osf.io/8md3h), except for the exploratory post hoc analyses (presented below).

### Ethics

The study was carried out under the housing and breeding permit no. 311.4-si (by Landratsamt Starnberg, Germany), which covers all implemented procedures, including blood sampling individuals for parentage assignment.

### Background of study populations and assortative mating

The zebra finches used in this study originate from 4 captive populations maintained at the Max Planck Institute for Ornithology: 2 domesticated (referred to as "Seewiesen" (S) and "Krakow" (K)) and 2 recently wild-derived populations ("Bielefeld" (B) and "Melbourne" (M)). For more background and general housing conditions, see [30,53,61]. The 4 populations have been maintained in separate aviaries (without visual and with limited auditory contact). When birds from 2 different populations (combining S with B and K with M) were brought together in the same breeding aviary, they formed social pairs that were predominantly assortative with regard to population (87% assortative pairs), despite the fact that opposite-sex individuals were unfamiliar with each other ([52]; see also [62]). To find out whether this assortative mating took place because of genetic (e.g., body size) or cultural (e.g., song) differences, we produced an offspring generation ("F1") in which half of the birds were cross-fostered between populations (between S and B or between K and M) and half of the birds were cross-fostered within populations. For this purpose, we used 16 aviaries (4 per population), each containing 8 males and 8 females of the same population that were allowed to freely form pairs and breed. Cross-fostering was carried out at the aviary level, such that 2 aviaries per population served for cross-fostering within population and the other 2 for between-population cross-fostering. This resulted in 8 cultural lines (4 populations × 2 song dialects), each maintained in 2 separate

aviaries (16 aviaries). When unfamiliar individuals of the 2 song dialects were brought together in equal numbers (50:50 sex ratio), they mated assortatively regarding song (79% assortative pairs [44]) but not regarding genetic population. To disentangle the song effect of interest from possible side effects of the cross-fostering per se, the 8 lines were bred for one more generation ("F2"). These F2 individuals are the focal subjects of this study. Breeding took place in 16 aviaries (2 per song dialect within population), but without cross-fostering. The 2 replicate aviaries of each song dialect line each contained 8 males and 8 females that produced the next generation. A subset of the resulting offspring ($n$ = 144, not used in this study, but see below) were used to test mate choice within each of the 4 genetic populations (here referred to as "F2 pilot experiment"). Again, we observed assortative pairing for song dialect (73% assortative pairs). The remaining F2 offspring were used as candidates for the experiment, as explained below.

## Experimental setup

To quantify the female fitness consequences of having preferences for males that are either rare or overabundant, we used 10 aviaries (3 for populations B and K and 2 for populations S and M). Each semi-outdoor aviary (measuring 4 m × 5 m × 2.5 m) contained 12 males and 12 females of the same genetic population, but from 2 different song dialects such that 4 females encountered 8 males of the same dialect, while the remaining 8 females encountered only 4 males of their own dialect. For each experimental aviary, we used individuals that were raised in 4 separate aviaries (2 of each song dialect) to ensure that opposite-sex individuals were unfamiliar to each other.

The allocation of birds to the aviaries followed 2 principles. First, we listed for each of the 16 rearing aviaries the number of available female and male F2 offspring that had not been used previously (in the "F2 pilot experiment") and that were apparently healthy (374 birds). Depending on the number of available birds, each rearing aviary was then designated to provide either 4 or 8 birds of either sex, such that the total number of experimental breeding aviaries that could be set up was maximized (10 aviaries). Second, the allocation of the available individuals within each rearing aviary to the designated groups of 4 or 8 individuals of a given sex was decided by Excel-generated random numbers. For example, if a given rearing aviary had 17 candidate female offspring, individuals were randomly allocated to a group of 4 for one experimental aviary, a group of 8 for another aviary, and a group of 5 as leftover (not used). This allocation procedure may have introduced a bias, because rearing aviaries that were highly productive (had more offspring) were more frequently designated to send groups of 8 offspring to an experimental aviary, while those that produced fewer offspring (in the extreme case fewer than 8 of one sex) were more likely used to send a group of 4 offspring to an experimental aviary. This might bias our fitness estimates if offspring production was partly heritable or if housing density prior to the experiment influenced the fitness in the experimental aviaries. We therefore assessed these potential biases in the statistical analysis (see Model 1b below).

After allocating individuals to the 10 experimental aviaries, 1 female (designated for aviary 2) and 2 males (designated for aviary 3) died before the start of the experiment. These individuals were then replaced by randomly choosing individuals of the same sex and rearing aviary, which, however, had previously taken part in the "F2 pilot experiment" (January 17 to 30, 2019). These replacement birds differed from the other individuals in the experiment, in that they had previous experience of nest building and egg laying >100 days before the start of experiment.

The 120 focal females had hatched in one of the 16 natal aviaries between May 30 and September 25, 2018 and remained in their natal aviaries initially together with their parents (which were removed between December 10, 2018 and January 16, 2019). On May 6, 2019, all

individuals used in the experiment were transferred to the 10 aviaries, whereby the 12 males and 12 females in each aviary were separated by an opaque divider. After 1 week, the divider was removed, and the experiment started. At this time, females were on average 313 days old (range: 230 to 348 days). To facilitate individual identification, each of the 12 males and 12 females within each aviary was randomly assigned 2 colored leg bands (using the following 12 combinations: blue–blue, black–black, orange–orange, orange–black, red–red, red–blue, red–black, white–white, white–black, white–orange, yellow–yellow, and yellow–blue).

## Breeding procedures

Each of the 10 experimental aviaries was equipped with 14 nest boxes. All nest boxes were checked daily during weekdays (Monday to Friday) for the presence of eggs or offspring. Eggs and offspring were individually marked, and a note was made whether eggs were warm. For eggs laid on weekends, we estimated the most likely laying date based on egg development. We collected a DNA sample from all fertilized eggs (including embryos and nestlings that died naturally), unless they disappeared before sampling (see below) to determine parentage. Eggs containing naturally died embryos ($N = 343$) were collected and replaced by plastic dummy eggs (on average $12 \pm 4$ (SD) days after laying and $7 \pm 4$ days after estimated embryo death). Eggs that remained cold (unincubated) for 10 days ($N = 7$ out of 1,399 eggs) were removed without replacement and were incubated artificially to identify parentage from embryonic tissue. During nest checks, we noted the identity of the parent(s) that attended the nest (based on color bands) to clarify nest ownership for all clutches that were incubated.

As the main response variable, we quantified the reproductive success ("fitness") of each female in each experimental aviary as the total number of genetic offspring produced that reached the age of 35 days (typical age of independence). All eggs laid within a period of 70 days (between May 13 and July 22, 2019; $N = 1,399$) were allowed to be reared to independence; eggs laid after this period were thrown away and replaced by plastic dummy eggs to terminate a breeding episode without too much disturbance.

Out of 1,399 eggs, 319 eggs failed (180 appeared infertile, 101 disappeared, 30 broke, 4 had insufficient DNA, 3 eggs showed only paternal alleles (androgenesis), and 1 sample was lost). For the remaining 1,080 eggs, we unambiguously assigned maternity and paternity based on 15 microsatellite markers (for details, see [41]). Of these 1,080 fertile eggs, 750 developed into nestlings and 556 into offspring that reached 35 days of age.

The 1,399 eggs were distributed over 289 clutches (allowing for laying gaps of maximally 4 days), of which 190 (1,022 eggs) were attended by a heterosexual pair (involving 106 unique pairs), 55 (120 eggs) remained unattended, 24 (120 eggs) were attended by a single female, 12 (41 eggs) were attended by a single male, 3 (36 eggs) were attended by a female–female pair, 2 (30 eggs) were attended by 2 males and 2 females, 2 (21 eggs) were attended by a trio with 2 females, and 1 (9 eggs) by a trio with 2 males. Data on nest attendance were used to define 106 heterosexual social pairs. However, these include cases of re-pairing and cases of polygamy, such that a total of 95 different males and 95 different females participated in these 106 social pair bonds.

We also quantified 2 additional response variables for every female, namely the latency to start laying eggs (in days since the start of the experiment, counting to the first recorded fertile egg, and ascribing a latency of 75 days to females without fertile eggs) and the total number of fertile eggs laid within the 70-day experimental period (both based on the 1,080 eggs with genetically confirmed maternity). The 319 failed eggs were not considered.

Over the course of the experiment (May 13 to July 22 for egg laying and until September 9 for rearing young to independence) 1 male and 2 females (all of the more abundant type within

their aviaries) died of natural causes (a male in aviary 5 on June 28, a female in aviary 6 on July 24, and a female in aviary 10 on August 22). Thus, following the preregistered protocol, no bird was excluded from the data analysis.

## Data analysis

Following previously used methods [30, 63], we calculated "relative fitness" for each female i as N * number of offspring of female i / total number of offspring of all N females in the aviary. This index has a mean of 1 for each aviary and accounts for fitness differences between the 4 populations (note that all birds within an aviary come from the same population). Latency to egg laying was $\log_{10}$-transformed before analysis to approach normality. To control for the effect of inbreeding on fitness, we calculated female inbreeding coefficients F from existing genetic pedigree data (using the R package "pedigree" V.1.4, [64]). All mixed-effect models were built with the R package "lme4" V1.1–26 [65] in R version 4.0.3 [66], and p-values were calculated using the R package "lmerTest" V3.1–3 [67]. Note that for Gaussian models (lmer function), "lmerTest" calculates p-values from t-values based on Satterthwaite degrees of freedom, while for binomial models (glmer function; see below), p-values are calculated from z-values assuming infinite degrees of freedom. To get a more conservative p-value for the latter models, we refitted those as Gaussian models and used the estimated Satterthwaite degrees of freedom to manually calculate conservative p-values from z-values of binomial models.

Table 1 lists all the statistical models that compare the 2 treatment groups. These comprise both preregistered models (1 to 3) and post hoc exploratory models (4 to 15). All models have the same basic structure comparing a fitness-related trait between the 2 treatment groups (120 rows of data representing 80 high-competition and 40 relaxed-competition females). Thus, we used mixed-effect models with Gaussian (Models 1 to 3 and 5 to 12) or binomial (Models 4 and 13 to 15) errors, with the fitness-related trait as the dependent variable, with treatment as the fixed effect of interest, with the female's inbreeding coefficient as a covariate, and with the experimental aviary (10 levels) and the natal aviary (16 levels) as random effects. The covariate "inbreeding coefficient" was mean centered to render the model's parameter estimates (especially the intercept) directly interpretable [68].

For preregistered Model 1, we ran 2 versions (1a and 1b). Because the dependent variable of this model is relative fitness, which was scaled within experimental aviaries, the model was designed without random effects (as a general linear Model, 1a). To control for possible influences of the natal environment and of the genetic F1 mother, we added 2 mean-centered, fixed-effect covariates (in version 1b): (1) the total number of F2 offspring in the natal aviary where the focal female was raised (ranging from 29 to 45 offspring across the 16 natal aviaries); and (2) the number of independent F2 offspring produced by the genetic mother (1 year earlier, also within a 70-day window for egg laying; mean: 5.7, range: 0 to 12, N = 66 mothers of the 120 focal females).

## Exploratory analyses

To quantify the extent of assortative mating with regard to song dialect at the behavioral level, we relied on the 106 unique heterosexual pairs that were observed caring for at least 1 of 190 clutches (comprising 1,022 eggs). For the quantification of assortment on the genetic level, we relied on the genetic parentage of the 1,080 successfully genotyped eggs, of which 6 eggs had to be excluded because they were sired by males from the females' natal aviaries (due to sperm storage, N = 4), because alleles from 2 males were detected (presumably due to polyspermy, N = 1) or because no paternal alleles were detected (possible case of parthenogenesis, N = 1), leaving 1,074 informative eggs.

For each female, we scored their social pairing behavior, i.e., we noted whether they had been recorded as a member of one of the 106 heterosexual pairs engaging in brood care (see above). We quantified (a) the total number of social bonds (0, 1, or 2); (b) the number of assortative and disassortative bonds; and (c) the latency to their first social bond (i.e., the laying date of the first egg in a clutch they attended as one of the 106 pairs, relative to the start of the experiment; ascribing a latency of 75 days to females with zero social bonds). Latency was $\log_{10}$-transformed before the analysis.

For each female, we also counted the number of clutches (0, 1, or 2) attended as a single mother and we quantified (a) the number of eggs (out of the 1,080 genetically assigned eggs) they actively cared for themselves (in whatever social constellation); (b) the number of eggs dumped into nests attended by other females (in whatever social constellation, "egg dumping in the strict sense"); and (c) the number of eggs dumped anywhere ("egg dumping in the wide sense", including in nests attended by single males and in unattended nest boxes). All exploratory mixed-effect models (4 to 15) closely follow the design of the preregistered Models 2 and 3 (see above).

Models 13 to 15 deal with proportions of eggs, and, hence, we used binomial models with counts of successes and failures and controlling for overdispersion by fitting female identity (120 levels) as another random effect.

Model 8 on the latency to the first egg attended as a social pair deals with partly censored data, because a considerable fraction of females (21%) were not recorded in a social pair and were assigned a latency of 75 days. We therefore ran an additional Cox proportional hazard model (Model 8a), which models the probability (conventionally referred to as risk or hazard) of pairing over the course of time. This model was built using the R package Coxme V2.2–16 [69]. Note that we did not run such a model for the latency to the first genetic egg (Model 3), because only 3 out of 120 females did not lay any eggs.

For post hoc exploration of subsets of the data that were not experimentally controlled (e.g., females of a certain pairing status), we generally used *t* tests for comparing group averages.

We ran exploratory analyses on the levels of extra-pair paternity of 84 females that were socially paired to only a single male (i.e., recorded with only 1 male, among the 106 nest-attending heterosexual pairs). These 84 females produced a total of 795 eggs with parentage information. However, we excluded 48 eggs (from 16 females) that were laid before the date of pairing of the focal female (genetic mother). Overall, 239 of the remaining 747 eggs (32%) were sired by a male that was not the social partner (the male with whom the female attended a nest), so these are classified as "extra-pair sired." We calculated levels of extra-pair paternity for 3 groups of females: (1) assortatively paired females of the relaxed-competition group (*n* = 35 females); (2) assortatively paired females of the high-competition group (*n* = 28 females, one of which did not lay any eggs with parentage information); and (3) disassortatively paired females of the high-competition group (*n* = 21 females). To compare levels of extra-pair paternity between the latter 2 groups of females, we used a *t* test on percentages of extra-pair paternity calculated for each female.

## Supporting information

**S1 Fig. Kaplan–Meier plot showing the time taken to social pairing under the 2 experimental treatments.**
(DOCX)

**S1 Table. Female relative fitness as a function of treatment and confounding factors.**
(DOCX)

**S2 Table. Number of genetically verified eggs laid per female as a function of treatment and female inbreeding coefficient.**
(DOCX)

**S3 Table. Latency (in days, log10-transformed) to lay the first genetically verified egg as a function of treatment and female inbreeding coefficient.**
(DOCX)

**S4 Table. Probability of remaining socially unpaired (not recorded participating in one of 106 nest-attending pairs) as a function of treatment and female inbreeding coefficient (binomial model on n = 120 females).**
(DOCX)

**S5 Table. Number of social pair bonds observed per female (range 0 to 2) as a function of treatment and female inbreeding coefficient (Gaussian mixed-effect model).**
(DOCX)

**S6 Table. Number of assortative social pair bonds observed per female (range 0 to 2) as a function of treatment and female inbreeding coefficient (Gaussian mixed-effect model).**
(DOCX)

**S7 Table. Number of disassortative social pair bonds observed per female (range 0 to 2) as a function of treatment and female inbreeding coefficient (Gaussian mixed-effect model).**
(DOCX)

**S8 Table. Latency (in days, log10-transformed) to the first recorded egg in a clutch attended as one of the 106 social pairs as a function of treatment and female inbreeding coefficient.**
(DOCX)

**S9 Table. Number of clutches attended as a single mother (range 0 to 2) as a function of treatment and female inbreeding coefficient (Gaussian mixed-effect model).**
(DOCX)

**S10 Table. Number of eggs that the female took care of (recorded as a social mother in any pairing constellation) as a function of treatment and female inbreeding coefficient.**
(DOCX)

**S11 Table. Number of genetically verified eggs by a female that were cared for by another female as a function of treatment and female inbreeding coefficient.**
(DOCX)

**S12 Table. Number of genetically verified eggs per female that she did not take care of as a function of treatment and female inbreeding coefficient.**
(DOCX)

**S13 Table. Relative counts of eggs that the female dumped (in a strict sense) versus her remaining eggs as a function of treatment and female inbreeding coefficient (binomial mixed-effect model).**
(DOCX)

**S14 Table. Relative counts of eggs that the female dumped (in a wide sense) versus took care of as a function of treatment and female inbreeding coefficient (binomial mixed-effect model).**
(DOCX)

**S15 Table. Relative counts of eggs that developed into independent young and her remaining eggs that did not reach independence as a function of treatment and female inbreeding coefficient (binomial mixed-effect model).**
(DOCX)

**S16 Table. Descriptive statistics on extra-pair mating by 3 groups of females (dependent on competition treatment and social pairing status).**
(DOCX)

**S17 Table. Cox proportional hazard model on the probability of social pairing over time (i.e., time to the first recorded egg in a clutch attended as one of the 106 social pairs) as a function of treatment and female inbreeding coefficient.**
(DOCX)

**S1 Text. R-code and model outputs of planned Models 1 to 3 and post hoc Models 4 to 15.**
(DOCX)

## Acknowledgments

We are grateful to M. Schneider for molecular work and to E. Bodendorfer, J. Didsbury, P. Neubauer, C. Scheicher, I. Schmelcher, and B. Wörle for animal care.

## Author Contributions

**Conceptualization:** Wolfgang Forstmeier, Daiping Wang, Bart Kempenaers.

**Data curation:** Wolfgang Forstmeier, Katrin Martin.

**Formal analysis:** Wolfgang Forstmeier.

**Funding acquisition:** Bart Kempenaers.

**Investigation:** Wolfgang Forstmeier, Daiping Wang, Katrin Martin.

**Methodology:** Wolfgang Forstmeier, Daiping Wang, Katrin Martin.

**Project administration:** Wolfgang Forstmeier.

**Resources:** Bart Kempenaers.

**Supervision:** Wolfgang Forstmeier, Bart Kempenaers.

**Validation:** Wolfgang Forstmeier, Katrin Martin.

**Visualization:** Wolfgang Forstmeier, Daiping Wang, Bart Kempenaers.

**Writing – original draft:** Wolfgang Forstmeier.

**Writing – review & editing:** Wolfgang Forstmeier, Daiping Wang, Katrin Martin, Bart Kempenaers.

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
