## [Editor Report · Decision Letter 0]

29 Apr 2021

Dear Dr Forstmeier, 

Thank you for submitting your manuscript entitled "Fitness costs of female choosiness in a socially monogamous songbird" for consideration as a Research Article by PLOS Biology.

Your manuscript has now been evaluated by the PLOS Biology editorial staff, as well as by an academic editor with relevant expertise, and I'm writing to let you know that we would like to send your submission out for external peer review.

Please re-submit your manuscript within two working days, i.e. by May 03 2021 11:59PM.

Kind regards,

Roli Roberts

Roland Roberts

Senior Editor

PLOS Biology

rroberts@plos.org

---

## [Decision Letter · Decision Letter 1]

16 Jun 2021

Dear Dr Forstmeier,

Thank you very much for submitting your manuscript "Fitness costs of female choosiness in a socially monogamous songbird" for consideration as a Research Article at PLOS Biology. Your manuscript has been evaluated by the PLOS Biology editors, an Academic Editor with relevant expertise, and by three independent reviewers.

You'll see that all three of the reviewers are very positive about your study, but each raises a number of concerns that will need to be addressed. The Academic Editor asked me to reveal his identity (Michael Jennions) and tell you that he has read the manuscript in the light of reviewer #1's comments and finds the logic of his argument "fairly compelling." While we will not absolutely insist that you follow this referee's presentational approach, Dr Jennions says "that it is worth thinking about it carefully rather than following the natural human tendency to stick with one's original set up. I find the logic of Ref 1's argument fairly compelling." Please attend to all points raised by the reviewers.

In light of the reviews (below), we are pleased to offer you the opportunity to address the comments from the reviewers in a revised version that we anticipate should not take you very long. We will then assess your revised manuscript and your response to the reviewers' comments and we may consult the reviewers again.

We expect to receive your revised manuscript within 1 month.

**IMPORTANT - SUBMITTING YOUR REVISION**

*Resubmission Checklist*

*Published Peer Review*

*PLOS Data Policy*

*Blot and Gel Data Policy*

Sincerely,

Roli Roberts

Roland Roberts

Senior Editor

PLOS Biology

rroberts@plos.org

REVIEWERS' COMMENTS:

Reviewer #1:

[identifies himself as Alexandre Courtiol]

I have read the important study from Forstmeier et al. which empirically explored an understudied aspect of mate choice: the role of mate availability. The paper is well written. The experimental design is simple and elegant. The statistics are suitable. The results are interesting and I salute the authors for having made the effort to pre-register the methods for this study on the Open Science Foundation website. Provided that the authors can successfully address my criticisms, and I am confident they can, I would consider this paper to represent a valuable contribution to the literature of mate choice and, more generally, to the fields of behavioural ecology and evolutionary ecology.

Main remarks:

My main reservation about the paper as it currently stands concerns its overall framing and main result. The authors claim that their experimental setup measures the cost of choosiness and that their results show that the cost is very small if not zero (e.g. lines 256-257). For reasons I will detail next, however, I don't think that the experimental setup actually measures the cost of choosiness, nor that the results show that the cost of choosiness is small. This removes little from the quality and merit of the study, but I think that a different angle should be used to present the exciting results on more solid ground. One option would be to focus on the impact of a change in mate availability.

The literature shows that measuring the cost of choosiness is rarely attempted, and I appreciate that the authors tried to do so. Even if I were willing to accept many practical caveats due to the inherent difficulty of the task, I am not, however, persuaded that the design of this particular study is appropriate for fulfilling this purpose. I will focus on four main issues.

First, I am not convinced by the overall logic of the experimental design: if two treatments are designed so as to influence a cost, then what do we learn about the cost when comparing the two treatments? At best the results directly reflect the artificial design: create very contrasted treatments and you could make the case for a large cost, and create very similar treatments and you could make the case for a small cost. Here we have two a priori very contrasted treatments but no cost, so what to conclude? Probably that the two treatments did not manipulate the cost as expected, not necessarily that there is no cost. In sum, measuring any cost like that tells us a lot about the study design but little about the biology. The authors may disagree with me but the general point is that they should justify why they think their experimental design allows for the measurement of the cost of choosiness. It would also be interesting, if they keep their framing as is, to relate the experimental conditions to the extent of the observed variation in mating availability found in nature.

Second, the measurement of the cost of choosiness should ideally depend on how choosiness is being defined. Perhaps it should even depend on how one thinks choosiness may have evolved. To give an extreme example, if one considers that cognitive structures have specifically evolved so as to allow individuals to discriminate more precisely between potential mates, then the cost of choosiness should include the cost of growing, maintaining and using such structures; and that would call for a very different type of experimental work than the one presented here. I am not arguing here that the brain is modular, nor that preferences may not have evolved by pure sensory biases, but the previous example illustrates a general problem to which there is an easy fix: the authors should make it very clear what they attempt to measure and what they cannot measure. This is important because that would help the reader to understand what exactly is zero. Probably not the overall cost of choosiness.

The particular cost the authors aimed at quantifying seems to be related to the decrease in mating rate associated with an increased choosiness. This is not a bad cost to measure and it has indeed been the focus of several theoretical papers, precisely because it is a general cost that may apply across species. Yet, this brings us to the third issue concerning the measurement of choosiness: the measurement of the cost of choosiness should ideally be done under conditions of fixed choosiness and benefits of choice. When everything varies, it is not clear how the cost of choosiness can be quantified, even in a model. This is why many models about the evolution of choosiness do not include plasticity. While these conditions of fixed choosiness and benefits are hardly ever met in practice, using metrics such as the number of offspring produced probably makes things worse since these integrative metrics capture, by definition, the whole cost/benefit balance. Perhaps a partial fix would be to also measure the cost using more direct consequences of the choice than the reproductive success, such as something related to the decrease in mating rate (the authors do measure the time to first egg, but why limit this to the first one?). Also, as mentioned above, this point calls for a precise justification about how and why the experimental design allows for the measurement of the cost of choosiness, especially in view of the fact that choosiness and the (perceived) benefits of the choice may vary between treatments.

Fourth --and this is my biggest reservation--, due to the general difficulty of measuring the cost of choosiness, studying this cost in conditions where it is likely to be high may be easier. The authors seem to agree with this, but I see two inter-related problems with the particular experimental conditions that the authors used. In these conditions, and perhaps in nature, the studied species, Zebra finches, live in high density and are highly promiscuous. It is not clear why anyone would expect an increase in choosiness to trigger a sharp decrease in mating rate in such conditions. In other words, the authors study the cost of choosiness in a species where we already have reasons to believe that it is low. Had they worked on spiders where most individuals die before having had the chance to meet a potential partner, the situation would be quite different. Surprisingly, the authors did seem to expect a high cost of choosiness in their species. In fact, the key premise of the study is that "Selection against choosiness should be strongest in socially monogamous mating systems" (from the abstract) because the mating system is shaping the costs of choosiness. I may agree with that if they would have dropped the word "socially", but that word is important. Indeed, if selection is measured via genetic contributions, then the costs and benefits of choosiness should be shaped by the genetic mating system, irrespective of the social one. Of course, whether birds do find a social mate does impact on their realised fitness. But it does so not through shaping the costs and benefits of choosiness for genetic mates; it does so through shaping the costs and benefits of choosiness for social mates. In sum, I see a discrepancy between the level at which the experimental design plays out (social level) and the level at which the cost of choosiness is being measured (number of eggs, i.e. genetic level). Due to this discrepancy, I fear that the study cannot actually "quantify the fitness costs of having mating preferences that are difficult to satisfy" (from the abstract), since indeed as the authors show, the preferences are not particularly difficult to satisfy at the genetic level. 

Perhaps this is a matter of semantics, but I think that there is more to it than that. The authors seem to regret that the theoretical literature disregards what females may do when their preferences are not satisfied. For instance, the authors go on to propose that females may opt for extra pair copulations. However, from the theoretical perspective of the models discussed (at least for the ones I know), there is no issue: if females can satisfy their preferences --whether directly, or via extra pair copulations-- then their preferences are precisely satisfied, and the question of what they do when they are not becomes moot.

To suggest a solution to what I have described, I propose to frame the study as investigating the impact of mate availability (which is indeed one important aspect related to choosiness). This would avoid falling into HARKing while still doing justice to the pre-registered methodology. I do want to emphasize that I think that the design is great and reveals very interesting information about the biology of mate choice. I just don't think that it measures the cost of choosiness in any meaningful way. I also vaguely thought of perhaps trying to decompose the cost further (i.e. disentangling the cost of choice for the social partner from that of the choice for the genetic partner), but the authors would know best what to do.

Do my criticisms imply that this study is weak? No. Not at all. This study reveals an interesting relationship between social and genetic pairings: when preferences are easy to satisfy socially, social and genetic pairings seem very much in line. When they are not, then the correlation between social and genetic pairing decreases, with the social choice becoming more random and the genetic choice remaining unaltered. I don't know the bird literature well enough to know if this is already well known and established, but I find this very interesting. Indeed, this result shows, at least in zebra finches and under these experimental conditions, that what matters is fulfilling one's preferences for song dialects at the genetic level. This result may give insights about the benefits related to the particular preferences for the trait. The decrease in the correlation between social and genetic pairings also shows that individuals are capable of plasticity that seems adaptive. Whether or not this reflects the result of past selection in the wild is an interesting question. One could speculate that such plasticity would have been selected for the particular reason that the correlation between genetic preferences and social mate availability varies substantially. Or perhaps, this reflects plasticity that would have evolved for more general purposes… It also begs the question of why resistance against brood parasitism has not evolved. I vaguely remember works from Bielefeld people or their colleagues that did show that kin recognition matters for parental care after hatching, perhaps this would be a counter strategy. Or perhaps tolerance is a kind of cooperation that would make sense in the natural conditions. In short, this study is very interesting, well done, and thought provoking. A small rephrasing of the introduction and the discussion would suffice to avoid all the aforementioned problems and turn this work into a great paper for Plos Biology.

Minor general remarks:

One interesting set of figures that could be added to the paper (or as SI) would be a comparison between social and genetic levels. In particular, one figure could represent the number/proportion of assortative and disassortative pairings within each experimental treatment comparing both for social and genetic partners. Unless I missed it, in the current presentation one cannot directly tell how strong the assortative mating is at the genetic level in each treatment. (The pooled numbers are given line 190 and social level in figure 4). I would also find it interesting to see the breakdown of relative fitness and egg dumping in this light: genetic vs social for each treatment. That is, to make the equivalent of figure 5 for the genetic level. That way it would be easier for the reader to directly compare the social and genetic levels throughout.

Statistics: the statistics are generally very good. I only have 2 small remarks:

The authors sometimes seem to switch from (G)LMMs to simple t-tests (e.g. line 227). It is not clear to me why they do so. I don't think they have justified it. If the switch is to account for heteroskedasticity, the authors may want to try using the R package spaMM, which would allow them to do everything lme4 can do, but also much more, including defining a model for the residual variance. The syntax would thus resemble something like: spaMM::fitme(response ~ trt + ... + (1|ExpAV) + (1|NatalAV), resid.model = ~ trt, data = dd, family = ...). The R package glmmTMB and its dispformula argument would be another alternative. I am not sure this is required here but spaMM would also allow one to define random effects acting on the residual variance (i.e. spaMM can fit DHGLM sensu Lee & Nelder).

I think authors could do a little better than "p-values were calculated from t-values assuming infinite degrees of freedom" (line 437), which is not a conservative approach. Perhaps the authors did not do that anyhow since I see corrected degrees of freedom in the tables. Yet, if anything the degrees of freedom should be approximated, it would probably be best to underestimate them instead of overestimating them since an increase in degrees of freedom leads to more false positives. Anyhow, there is no need to reinvent the wheel since it has become straightforward these days to test the effect of variables in (G)LMM using parametric bootstrap (or applying a Kenward-Roger approximation, although I don't recall if that latter approach is appropriate for GLMM too). This is true irrespective of using lme4, spaMM or glmmTMB since some of these packages directly offer these options (e.g. spaMM), or have companion packages developed for such purpose (e.g. the R package pbkrtest handles lme4 fits, and others exist too).

Line-by-line remarks:

Line 59-61: I would appreciate it if the authors would illustrate their statement with a few names of species where indeed most or all females mate with the same lekking male.

Line 71: my reading of these papers is that the indirect benefits are small but not that the costs are small; if the authors agree with me, they should rephrase their statement.

Line 100: "As a consequence, such preferences will be strongly selected against" -> "can be strongly selected against". Indeed, it all depends on the difference in reproductive success between females that did meet their preferences and those that did not. If the difference is large enough, it won't be selected against. Which is why the authors correctly continued with "particularly when a male ornament is a poor indicator...".

Lines 136-137: it is not clear in this paragraph if the 4 vs 8 / 8 vs 4 is just an example or the only two experimental conditions used in the study. We later learn the latter interpretation is correct.

Line 158: it is indicated that 556 offsprings reached independence and figure 3 indicates that 1074 eggs were fertilized. The authors should mention and explain the large discrepancy between these 2 numbers.

Line 169 and elsewhere: the authors indicate an interquartile range but the distribution on which such range applies is not specified. Is that the interquartile range of raw data? Of the back-transformed means? Of something else?

Line 173: I would not write "reduce" here but instead speak of cost avoidance.

Lines 187-188: the authors should perhaps clarify what the maximum achievable number of assortative pairs is (80 if I got it right, which does illustrate that 72 is very close to expectations under perfect assortative mating).

Line 208: authors should add the degrees of freedom for the results cited that are not in a table.

Line 215: authors should recall the p-value of the non-significant results, as they did for significant ones.

Lines 211-223: authors could try to provide effect sizes in the text or illustration as figures for readers to get a sense of how different females from both groups behave without having to dig into SI tables.

Lines 256-257 + 301-302: I disagree (see main comments above).

Line 307: indeed it would be good if the authors discuss the differences between the results they show here and those they published in ref 25. One important aspect is how the benefits of choice are being influenced in the two studies.

Figure 1: the y-axis should not be labelled "relative female fitness", since, by definition, the mean relative female fitness should always be 1. I would perhaps remove the 1, keep the 0 and label the axis as "female fitness".

Figure 2: perhaps the authors should indicate some more information for the non r-iens: that data have been jittered horizontally for visual purposes, that those are boxplots, what whiskers represent… They did however mention that the horizontal bars indicate means rather than the (default) median, which is the most important detail since everything else is quite standard. (Same for fig 5).

Figure 4: the authors may want to revise their choice of colour so that B&W printing works too.

Alexandre Courtiol

Reviewer #2:

[identifies himself as Pr François-Xavier Dechaume-Moncharmont]

This manuscript deals with an important and largely underexplored question, the actual fitness cost of choosiness during pair formation. The experimental study is well designed using a very large dataset (8 cultural lines, 120 focal females, 289 clutches, 1399 eggs, 556 offspring over 4 months experiments). These results are highly valuable, and they will surely interest large audience readership. I also generally appreciated the thorough statistical analyses and the clarity of the methodological description about these questions in the Material and Methods section. The authors are fully transparent about the a priori and post-hoc hypotheses (through pre-registered protocol). They took great care to confirm genetics maternity and to consider inbreeding in the analyses. My overall impression about this manuscript is thus largely positive. I recommend acceptation after revisions because some points still deserve clarification in the methods, analysis and discussion.

Major comments

A major limitation of the present is that male traits submitted to female preference is a song dialect. The authors have extended and recognized expertise on this question: in previous works and in the present MS (preliminary tests), they provided convincing evidence that there is assortative pairing for dialect in this species. They also prudently try to explain the adaptive significance of such preference in the discussion (Lines 283 and followings). Yet, the crucial points here is that this trait is almost costless for the male, and is not genetically inherited (the song repertoire has been learned by imitation during ontogeny). Therefore, it does not fully satisfy two important components of sexual selection. (1) Honest signal: it makes sense for the female to choose a male based on a given traits if this trait is correlated with either the fitness quality of the male or the (direct or indirect) fitness gain for the female. (2) Heritability: to allow co-evolution between signal in males and preference in females, the male trait should be heritable. In the present study, the heritability of the male dialect is extremely weak, and only by cultural transmission (as nicely illustrated by the cross-fostering design in preliminary experiments), and the correlation between male trait and potential male quality is weakly discussed, and only as possible explanation in the discussion. This could be an issue as this study investigates the fitness gain arising from partner choice in absence of competition, or its symmetric, the fitness cost arising from absence of choice under competition. If the male's dialect does not indicate its quality, one could not expect large effects of female choice on her fitness. I recommend that the authors address this possible limitation more explicitly and earlier in the MS.

Minor comments

Lines 101, 109 and 138. The authors refer to a game theory process of "negative frequency dependence". The associated references (34-37) do not help to see why they use this concept here. The definition given lines 101-103 is incorrect. "Negative frequency dependence" refers to situation in which rare morphs (or strategies) are favoured compared to more frequent morphs. I do not understand its relevance in the cases discussed in the present MS. Clarification or rewriting are requested here.

While I acknowledge the attempt of clarification of the several fitness costs of choosiness in females (Fig. 1), I was highly disturbed by the wording "cost of dissatisfaction" because this arrow can also represent the cost due to poor male quality. This arrow aggregates both opportunity costs for the female and costs due to bad male quality. Would it be possible to use a less anthropomorphic wording choice?

Statistical analysis. Several models used scaling of covariates to report effect size measures (for instance line 711), but this scaling only subtracted the mean and did not divide by the standard deviation as illustrated by the corresponding R code line 812 in the parameter of the function scale(…, scale = FALSE). Could the authors justify this choice? If they aim at providing standardized effect sizes to allow comparisons (within or between studies), I consider that full scaling (mean and variance) is more appropriate.

Relative fitness gain. I was also disturbed by the calculation of relative fitness gain. While the provided equation (line 422) centres the average value on 1 for each aviary, it does not control for unequal variance between aviary. I consider that relative fitness gain for female i should be calculated as scaled metrics : RF_i = (F_i - mu)/sd, were F_i was the actual fitness of female i, mu is the mean fitness of the female in the aviary and sd the standard deviation of the fitness in the aviary. Using this scaling transformation, the mean RF in each aviary is the same (equal to zero) and the variance is equal to 1. The authors should justify their calculation choice focusing only on mean scaling and not mean and variance scaling.

Lines 202 and followings. There is a far better way to cope with censored data (when the female did not lay eggs) than exclude them from the analysis. Cox proportional hazard model are design for that very purpose, with time data. Instead of dumping the information that a female has not yet lay egg, it fully uses the information that no eggs have been laid at the end of the observation period. These models are easy to implement in R and their analyses are straightforward. I recommend that the authors analyse all times latency data as censored variables, bounded by the end of the experimental period using Cox models.

Figure 4 A and B, such pie charts are poorly informative and should be avoided. It's easy to replace them by two quick sentences in the main text.

Line 292. The reference is missing in the reference list.

--

Pr François-Xavier Dechaume-Moncharmont

University of Lyon, France

fx.dechaume@univ-lyon1.fr

Reviewer #3:

This study by Forstmeier et al. estimates the costs of female choosiness in the context of mate choice in captive populations of zebra finches. Estimating the costs of female preferences is important because such costs play a large role in determining expected outcomes in models of sexual selection by mate choice. I was quite impressed by this study, which is based on an imaginative design that exploits the finches' preference for 'local' (i.e. natal) song dialects. A clever cross-fostering design was also employed to avoid many potential statistical issues. The study was based on a pre-registered protocol. The authors did not find the result they expected (females that could not meet their preferences did not suffer an overall fitness cost) and they consequently also include exploratory (non-registered) analyses designed to uncover the reasons for this surprising 'non-result'. I appreciated the authors' transparency concerning the discrepancies between the pre-registered analyses and the current study and think the deviations are entirely reasonable and justified by the data. Their major unexpected finding was that females who remained unpaired (presumably due to the lack of preferred males) made up for this by dumping eggs in other birds' nests.

I have little in the way of major comments, as I found this article rigorous and well-written.

1. Socially monogamous and lekking systems differs not only in the costs of female choose (potentially higher under monogamy) but also potentially in the benefits (e.g. if there is large variation in direct benefits such as parental care provided by males). In my view, this could have explained more clearly in the introduction. 

2. In the discussion, I believe it would be helpful to speculate on the similarities/differences between aviary and wild conditions for this species. E.g. I'm aware that zebra finches often nest colonially in the wild - is the opportunity for egg dumping similar or much higher under aviary conditions? This is relevant to estimating the costs of choice during the species' evolutionary history (especially for the 'recently wild' lines).

3. The authors assumed infinite degrees of freedom for the t-tests based on mixed models. I'm aware that obtaining p-values from mixed models is a controversial topic and I am by no means an expert. My understanding, however, is that it is unwarranted (and anti-conservative) to assume infinite df in cases like this with a relatively small number of groups. I would suggest trying to roughly estimate the degrees of freedom based on similar fixed designs or using another approach (e.g. parametric bootstrap). If the original assumption is retained, it should be justified more rigorously.

4. One potential experimental flaw is already revealed (with refreshing transparency, I might add) by the authors:

This allocation procedure may have introduced a bias, because rearing aviaries that were highly productive (had more offspring) were more frequently designated to send groups of 8 offspring to an experimental aviary, while those that produced fewer offspring (in the extreme case fewer than 8 of one sex) were more likely used to send a group of 4 offspring to an experimental aviary. This might bias our fitness estimates if offspring production was partly heritable or if housing density prior to the experiment influenced the fitness in the experimental aviaries.

I think the authors did a good enough job of accounting for this potential source of error. However, I think they should refer to it in the discussion as well as the methods.

Otherwise I have only have minor comments:

L28: 'Selection against choosiness should be strongest in socially monogamous mating systems': This statement only applies to the choice of social mate, not to extra-pair mates.

L52: I would say 'fixed optimal level', as plastic responses can also be optimal

L54-5: Please add some citations to the huge theoretical literature on this topic. E.g.

Iwasa & Pomiankowski (1999). Good Parent and Good Genes Models of Handicap Evolution. (doi: 10.1006/jtbi.1999.0979)

Kokko et al. (2015). Mate-sampling costs and sexy sons. (10.1111/jeb.12532)

L59: 'The most spectacular examples of sexually selected traits have been observed in lek mating systems with strong reproductive skew'. The word 'spectacular' is subjective, of course, but I would not make this claim. Many paradigmatic examples (e.g. some birds-of-paradise) are not lekking species. 'Many of the most' would be better in my view.

L74: I would write 'more tractable' rather than 'better'. Monogamy with direct benefits is quite a different scenario from a theoretical perspective and the questions answered by studying monogamous and lek systems are not necessarily the same

L94-99: Another possible cost arises if non-preferred partners provide lower direct benefits (e.g. are worse parents). Also, this sentence is very difficult to read - I suggest rephrasing.

L105: This point is made very clearly by Fitzpatrick and Servedio (2018, doi: 10.1093/cz/zoy029) in the context of male mate choice.

L113: 'relatively little consensus': or simply a lack of strong preferences

L224-228: It's not clear which comparisons these t-tests represent.

L283: 'assortative mating was present … at the genetic level': I puzzled over this for a while, because song dialect is not genetically determined. I now realize you meant to contrast genetic paternity with social pairing. I suggest rephrasing this so that it's easier to digest.

L297-298: 'whereby we predicted that these costs' -> 'which we predicted'

L299: 'Our study did not fail in the sense that our treatment was effective': I suggest framing more positively, e.g. 'Although our experimental treatment was effective in eliciting strong assortative mating preferences'

Typos etc.

L43: 'equal fitness as' -> 'equal fitness to'. Similarly on L229.

L95: delete comma before 'can'

L141: 'two-third majority' -> 'two-thirds majority'

L154: delete 'part of' (it reads weird and it's clear that only part of the analysis was unplanned)

L246: 'inform us about'. Other alternatives (perhaps preferable): 'inform discussions about' or 'help frame discussions about'

L305: delete comma before 'were'

L309: 'models on' -> 'models of'

L353: Does the 'Hence' belong here?

L393: 'naturally died embryos and nestlings' -> 'embryos and nestlings that died naturally' [or 'of natural causes']

L393: 'naturally died' -> 'dead'? Or rephrase as above.

Fig. 3: get rid of the red squiggles from Word :-)

Fig. 5 legend: 'wide' -> 'broad' for consistency

---

## [Editor Report · Decision Letter 2]

16 Sep 2021

Dear Dr Forstmeier,

Thank you for submitting your revised Research Article entitled "Fitness costs of female choosiness in a socially monogamous songbird" for publication in PLOS Biology. The Academic Editor and I have assessed your revisions and responses to the reviewers' comments.

Based on this assessment, we will probably accept this manuscript for publication, provided you satisfactorily address the remaining points raised by the reviewers. Please also make sure to address the following data and other policy-related requests.

IMPORTANT:

a) We wonder if you could provide a more informative and declarative title. We usually prefer titles that contain an active verb and which contain no punctuation. Something along the lines of "Socially monogamous females avoid costs of choosiness by XYZ..." or "Female choosiness in a socially monogamous bird incurs XYZ costs...." might do, but I do realise that the findings are complex. I'm happy to discuss this further if you like.

b) Many thanks for providing the underlying data in OSF. Please could you indicate the location of these data clearly in each relevant main and supplementary Figure legend? e.g. "The data underlying this Figure may be found in https://osf.io/6e8np/"

We expect to receive your revised manuscript within two weeks. 

*Published Peer Review History*

*Early Version*

Sincerely,

Roli Roberts

Senior Editor,

rroberts@plos.org,

PLOS Biology

DATA NOT SHOWN?

---

## [Editor Report · Decision Letter 3]

7 Oct 2021

Dear Dr Forstmeier,

On behalf of my colleagues and the Academic Editor, Michael Jennions, I'm pleased to say that we can in principle offer to publish your Research Article "Fitness costs of female choosiness are low in a socially monogamous songbird" in PLOS Biology, provided you address any remaining formatting and reporting issues. These will be detailed in an email that will follow this letter and that you will usually receive within 2-3 business days, during which time no action is required from you. Please note that we will not be able to formally accept your manuscript and schedule it for publication until you have made the required changes.

IMPORTANT: Please note that I have changed the title of the manuscript to the active form that you suggested in your cover letter.

PRESS: We frequently collaborate with press offices. If your institution or institutions have a press office, please notify them about your upcoming paper at this point, to enable them to help maximise its impact. If the press office is planning to promote your findings, we would be grateful if they could coordinate with biologypress@plos.org. If you have not yet opted out of the early version process, we ask that you notify us immediately of any press plans so that we may do so on your behalf.

Best wishes, 

Roli Roberts

Roland G Roberts, PhD 

Senior Editor 

PLOS Biology

rroberts@plos.org